# TS-SNN: Temporal Shift Module for Spiking Neural Networks

**Kairong Yu** [* 1]   **Tianqing Zhang** [* 1]   **Qi Xu** [2]   **Gang Pan** [1]   **Hongwei Wang** [1]

## Abstract

Spiking Neural Networks (SNNs) are increasingly recognized for their biological plausibility and energy efficiency, positioning them as strong alternatives to Artificial Neural Networks (ANNs) in neuromorphic computing applications. SNNs inherently process temporal information by leveraging the precise timing of spikes, but balancing temporal feature utilization with low energy consumption remains a challenge. In this work, we introduce Temporal Shift module for Spiking Neural Networks (TS-SNN), which incorporates a novel Temporal Shift (TS) module to integrate past, present, and future spike features within a single timestep via a simple yet effective shift operation. A residual combination method prevents information loss by integrating shifted and original features. The TS module is lightweight, requiring only one additional learnable parameter, and can be seamlessly integrated into existing architectures with minimal additional computational cost. TS-SNN achieves state-of-the-art performance on benchmarks like CIFAR-10 (96.72%), CIFAR-100 (80.28%), and ImageNet (70.61%) with fewer timesteps, while maintaining low energy consumption. This work marks a significant step forward in developing efficient and accurate SNN architectures.

## 1. Introduction

Spiking Neural Networks (SNNs) have emerged as a promising alternative to traditional Artificial Neural Networks (ANNs) due to their biological plausibility and energy efficiency. SNNs emulate the behavior of biological neurons, which communicate through discrete spikes rather than continuous values. This spiking mechanism allows SNNs to operate with significantly lower power consumption, making them ideal for applications in neuromorphic computing and edge devices where energy efficiency is crucial(Yamazaki et al., 2022; Taherkhani et al., 2020). In comparison to ANNs, which process information in a static and continuous manner, SNNs inherently incorporate temporal dynamics by encoding information in the timing of spikes. Although SNNs are biologically inspired neural networks, there is still a significant gap between current SNN models and the actual biological neural networks they aim to emulate. Neuromodulators like dopamine and norepinephrine are released in response to previous neural activity and modify neuronal excitability and synaptic strength, influencing current and future activity (Turrigiano & Nelson, 2004; Alcamí & Pereda, 2019). Additionally, neurons adapt through mechanisms such as frequency-dependent synaptic plasticity and ion channel regulation, which allow them to respond based on their activity history and present status (Turrigiano & Nelson, 2004; Alcamí & Pereda, 2019). These mechanisms ensure that synaptic transmission is influenced by both past and future activity, enabling dynamic adjustments in the nervous system. Temporal modeling can be employed to handle the temporal information mentioned above. It is a well-established technique in sequential data processing. However, due to the inherent temporal characteristics of SNNs, there is an overlap with video signal processing. This suggests that temporal modeling techniques used in other domains, such as video recognition, can be effectively applied to SNNs. Although temporal modeling in SNNs is promising, effectively leveraging temporal information while maintaining low energy consumption remains a significant challenge. Previous methods have struggled to balance these two aspects. A major issue is that many existing methods either fail to fully exploit the temporal dimension or introduce substantial computational costs, negating the energy efficiency advantages of SNNs.

To address this challenge, we propose a novel approach that integrates temporal shift operations into SNNs, forming Temporal Shift module for Spiking Neural Networks (TS-SNN). The principle of the TS module is to distort spike features along the temporal dimension through a shift operation. This enhances the utilization of temporal information in SNNs, reducing the forgetting of past timestep information and improving the learning of future timestep information, all while introducing only minimal computa-

---
[*]Equal contribution [1]Zhejiang University [2]Dalian University of Technology. Correspondence to: Hongwei Wang <hongwei-wang@intl.zju.edu.cn>, Qi Xu <xuqi@dlut.edu.cn>.

*Proceedings of the 42$^{nd}$ International Conference on Machine Learning*, Vancouver, Canada. PMLR 267, 2025. Copyright 2025 by the author(s).

tional cost. By integrating spiking features across different timesteps, TS-SNN effectively mitigates the impact of initial timestep information loss and addresses the issue of insufficient feature extraction at end timesteps. This strengthens long-term temporal dependencies, thereby improving the accuracy of SNNs in related tasks. Our main contributions are summarized as follows:

- We propose a Temporal Shift Module to address the inherent limitations in the temporal dynamics of traditional SNNs. This module efficiently integrates past, present, and future spike features through a simple shift operation, enhancing SNNs' temporal modeling capabilities with minimal additional computation. The TS module is easily incorporated into any SNN architecture.

- We introduces a residual combination method for the TS module, which integrates shifted and original features to prevent information loss after shifting. A penalty factor is utilized to maintain training stability, thereby enhancing the network's ability to capture temporal dynamics while preserving original features.

- A series of extensive ablation studies were conducted to optimize the TS module and gain deeper insights into its effectiveness. The performance of TS-SNN was then evaluated on various benchmark datasets, demonstrating state-of-the-art results with fewer timesteps. Additionally, computational energy consumption analyses were performed to verify the energy efficiency of TS-SNN.

## 2. Related Works

### 2.1. Training of Deep SNNs

SNN training typically follows two approaches: ANN-to-SNN conversion and direct SNN training. The former conversion methods enable the utilization of existing ANN models has been made a lot works (Han & Roy, 2020; Wang et al., 2023a) but have limitations in fully leveraging the spatio-temporal information intrinsic to SNNs due to the inherent absence of this dimension in ANNs (Hu et al., 2024). Direct SNN training methods, particularly unsupervised learning approaches like Hebbian learning(Hebb, 2005) and Spike-Timing-Dependent Plasticity (STDP)(Bi & Poo, 1998), are limited in scalability for deep networks and large datasets. To address this, (Chakraborty et al., 2021) proposed a hybrid model combining unsupervised STDP with backpropagation for energy-efficient object detection.

Directly learning SNNs is challenging due to the undifferentiable nature of spike firing. The surrogate gradient (SG) approach addresses this issue, thereby enhancing the practicality and applicability of direct SNN training (Neftci et al., 2019; Lee et al., 2020; Wu et al., 2019; Fang et al., 2021a). Recent advancements in SNN architectures have introduced several improvement techniques. MPBN approach (Guo et al., 2023c) incorporates an additional Batch Normalization layer after membrane potential updates to stabilize training processes, DA-LIF (Zhang et al., 2025a) introduce an additional learnable decay on conventional LIF. (Duan et al., 2022) proposed the TEBN method, leveraging distinct weights at each timestep to enhance learning dynamics, RSNN (Xu et al., 2024) enhanced spike-based data processing in SNNs. SEW-ResNet (Fang et al., 2021a) introduced a widely utilized ResNet backbone for SNNs, MS-ResNet (Hu et al., 2024) reorganizes the construction of Vanilla ResNet layers. (Zheng et al., 2023) explored spike-based motion estimation for object tracking through bio-inspired unsupervised learning. (Doutsi et al., 2021) introduced a dynamic image quantization mechanism enhancing visual perception quality over time by leveraging both time-SIM and rate-SIM for encoding spike trains. (Xu et al., 2023a;b; Yu et al., 2025a) proposed knowledge distillation methods to train deep SNNs using ANNs as the teacher model and SNNs as the student model. These innovations collectively contribute to the advancement of SNNs, making them more practical and effective for a wider range of applications. While existing research addresses various efficient method to enhance SNNs, the survey indicates the absence of an approach fuse features across different timesteps in SNNs.

### 2.2. Temporal Shift Modeling

In deep learning, temporal modeling refers to the process of analyzing based on sequential data that evolves over time where the order and timing of data points are important. Due to the intrinsic temporal characteristics of SNNs, there is a natural overlap with video signal processing, suggesting that temporal modeling techniques used in other domains, such as video recognition, can be effectively applied to SNNs. Among traditional temporal modeling approaches, 3D CNNs are the most straightforward method(Guo et al., 2020; 2023a). While they can directly process spatiotemporal features, but they come with a high computational cost. To address this, (Lin et al., 2019) proposed TSM, which transforms 3D CNNs into 2D CNNs by shifting a small portion of channels along the temporal dimension, thereby achieving the performance of 3D CNNs at the cost of 2D CNNs. Inspired by the principles behind TSM, ACTION-Net (Wang et al., 2021) introduced a three-path architecture for enhanced action recognition, while the Learnable Gated Temporal Shift Module (LGTSM) (Chang et al., 2019) incorporated learnable kernels to improve temporal fusion efficiency. (Wu et al., 2018a) combined the shift operation with $1 \times 1$ convolutions as an efficient alternative to $3 \times 3$ convolutions. Further advancements include the proposal of learnable active shifts by researchers (Chen et al., 2019; Jeon

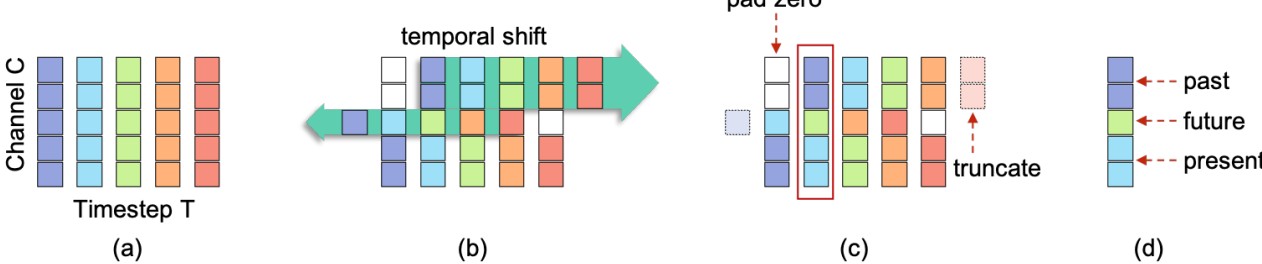

*Figure 1.* **Illustration of Temporal Shift. (a)** Initial spike feature representation, where each row represents a different channel and each column represents a different timestep. **(b)** Temporal shift operation. Some channel are shifted forward (to the right) along the temporal dimension, while others are shifted backward (to the left), with the remaining features left unchanged. **(c)** To maintain consistent tensor dimensions after the shift, the empty spaces created by the shift are padded with zeros and excess values are truncated. **(d)** The shifted feature provides a fusion of past, present, and future information within each channel.

& Kim, 2018), and (Liu et al., 2021; Zhang et al., 2025b) implemented a self-attention mechanism through shifting windows to enhance the performance of vision transformers. (Yu et al., 2022) combined spatial shifts with multilayer perceptrons to achieve competitive performance in high-level vision tasks. (Yu et al., 2025b) proposes a spatiotemporal module leveraging frequency characteristics to enhance network performance. Additionally, (Li et al., 2023) extended the shift operation to grouped spatial-temporal shifts, demonstrating its applicability to video restoration tasks. These advancements in related studies demonstrate the necessity and effectiveness of temporal modeling. However, current research on SNNs on this domain is still quite limited. Our proposed method aims to fill this gap.

## 3. Methodology

In this section, we first provide a brief overview of the Leaky Integrate-and-Fire (LIF) model and the training process as discussed. Following that, we introduce the proposed TS module. Lastly, we described how to implement the TS module in SNN.

### 3.1. Preliminary of the Leaky Integrate-and-Fire Model and SNN Deep Training

The spiking neuron is a fundamental component of SNNs and the Leaky Integrate-and-Fire (LIF) neuron is employed due to its efficiency and simplicity. Mathematically, the discrete-time and iterative representation of LIF neuron are described as follows:

$$V^{t,n} = f(H^{t-1,n}, X^{t,n}), \tag{1a}$$
$$S^{t,n} = \Theta(V^{t,n} - v_{\text{th}}), \tag{1b}$$
$$H^{t,n} = V_{\text{reset}} \cdot S^{t,n} + V^{t,n} \odot (1 - S^{t,n}). \tag{1c}$$

The Heaviside step function $\Theta$ is defined as:

$$\Theta(x) = \begin{cases} 0 & \text{if } x < 0 \\ 1 & \text{if } x \geq 0. \end{cases} \tag{2}$$

Where $H^{t-1,n}$ represents the membrane potential after a spike trigger at the previous timestep, $X^{t,n}$ is the input feature of the $n$-th layer at timestep $t$. while $V^{t,n}$ denote the integrated membrane potential. $v_{\text{th}}$ is the firing threshold to determine whether the spike $S^{t,n}$ triggered.

During the network training of SNN, we utilize a spatial-temporal backpropagation (STBP) (Wu et al., 2018b) algorithm. The independently descriptions of temporal and spatial aspects are as follows:

$$\frac{\partial L}{\partial S_l^{t,i}} = \sum_{j=1}^{n^{l+1}} \frac{\partial L}{\partial S_{l+1}^{t,j}} \frac{\partial S_{l+1}^{t,j}}{\partial S_l^{t,i}} + \frac{\partial L}{\partial S_l^{t+1,j}} \frac{\partial S_l^{t+1,j}}{\partial S_l^{t,j}}$$
$$\frac{\partial L}{\partial V_l^{t,i}} = \frac{\partial L}{\partial S_l^{t,i}} \frac{\partial S_l^{t,i}}{\partial V_l^{t,i}} + \frac{\partial L}{\partial S_l^{t+1,i}} \frac{\partial S_l^{t+1,i}}{\partial V_l^{t,i}}. \tag{3}$$

Although the Heaviside Function 2 is non-differentiable, previous work has successfully overcome this through the surrogate gradients (SG) method (Neftci et al., 2019):

$$\frac{\partial S}{\partial V} = \frac{1}{a} \cdot sign(|V - V_{th}| < \frac{a}{2}), \tag{4}$$

where both $a$ and $V_{th}$ are hyperparameters usually set to 1. The gradient takes the value of 1 when $V$ is within the range of -0.5 to 0.5, otherwise it takes the value of 0.

### 3.2. Temporal Shift Module

The TS module enhances temporal dynamics in SNNs by shifting portions of the feature map along the temporal dimension. It divides the input feature tensor into segments and shifts them: some forward, some backward, and others

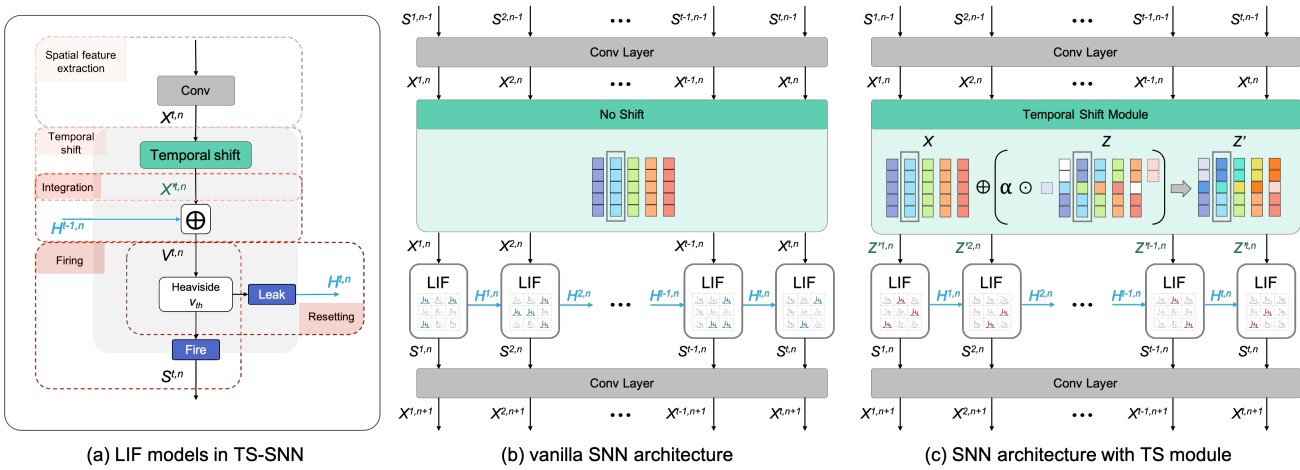

*Figure 2.* Overview of the TS-SNN Architecture.

unshifted, representing shifts of +1, -1, and 0 timesteps, respectively.

The TS module is designed to efficiently model temporal dynamics within SNN by shifting portions of the feature map along the temporal dimension. The Temporal Shift operation splits the input feature tensor into multiple segments and then assigns two random split points to divide the original features into three parts. These segments are then shifted along the temporal dimension: some forward, some backward, and others unshifted, representing shifts of +1, -1, and 0 timesteps, respectively.

Formally, for an input tensor $X \in \mathbb{R}^{T \times C \times H \times W}$, where $T$ is the number of timesteps, $C$ is the number of channels, $H$ is the height, and $W$ is the width. First, $C$ channels are divided into $C_k$ feature segments, each containing $C_{fold}$ channels, defined as:

$$C_{fold} = C/C_k. \tag{5}$$

Next, the tensor $Z$ after the temporal shift is defined as:

$$\begin{cases} Z^{+1} = X[t+1, :g_1 \cdot C_{fold}, :, :] \\ Z^{-1} = X[t-1, g_1 \cdot C_{fold} : g_2 \cdot C_{fold}, :, :] \\ Z^0 = X[t, g_2 \cdot C_{fold} :, :, :] \end{cases} \tag{6}$$

Where $g_1$ and $g_2$ represent the indices of the feature segments after random split, with $0 < g_1 < g_2 < C_k$ and $g_1 \neq g_2$. In Figure 1(a), the initial input spike feature maps with C channels and T timesteps are depicted. In Figure 1(b), some channels are shifted forward by an operation of +1, others backward by an operation of -1, while the rest remain unshifted along the temporal dimension demonstrating the process of Equation (6). However, this shift operation can result in vacancies or excesses at the edge timesteps where features are shifted away or out. To maintain consistent tensor dimensions after the shift operation, vacancies are padded with zeros, and excess values

---

**Algorithm 1** Temporal Shift Module

**Input:** Channel Folding Factor $C_k$; Input tensor $X$ of shape $[T, C, H, W]$, where:
$\quad T$: Timesteps, $C$: Channels, $H$: Height, $W$: Width.
**Output:** Shifted tensor $Z$.
**Method:**
Compute fold channel size $C_{fold} = C/C_k$.
Generate random indices $I_f$ for channel groups.
Determine $g_1$ and $g_2$ from $I_f$:
$\quad g_1 \leftarrow \min(I_f[0], I_f[1]), \ g_2 \leftarrow \max(I_f[0], I_f[1]).$
Initialize a zero tensor $Z$ with the same shape as $X$.
Shift a portion of $X$ left:
$\quad Z[:-1, :g_1 \times C_{fold}, :, :] \leftarrow X[1:, :g_1 \times C_{fold}, :, :].$
Shift a portion of $X$ right:
$\quad Z[1:, g_1 \times C_{fold} : g_2 \times C_{fold}, :, :] \leftarrow X[:-1, g_1 \times C_{fold} : g_2 \times C_{fold}, :, :].$
Leave the remaining portion of $X$ unchanged:
$\quad Z[:, g_2 \times C_{fold} :, :, :] \leftarrow X[:, g_2 \times C_{fold} :, :, :].$
Reshape $Z$ back to $[T, C, H, W]$.
**return** shifted tensor $Z$.

---

are truncated as shown in Figure 1(c). Figure 1(d) illustrates the multi-timestep fused feature matrix after the shift, indicating how the "past" (previous timesteps), "present" (current timestep), and "future" (upcoming timesteps) are fused within a single timestep. This fusion forms a complete data representation by stacking partial features from different timesteps. This TS operation can be performed without any multiplications or additions, thereby introducing no additional computational burden. Compared to traditional 2D convolution operations, the TS module achieves the most straightforward form of feature fusion, which aligns well with the energy-efficient and low-power nature of SNNs. The detailed Temporal Shift procedure for the TS-SNN is given in Algorithm 1.

## 3.3. Implementation of the TS Module

To integrate the TS module effectively into the network architecture, a common approach is to directly insert the module in place. However, this straightforward method can hinder the model's ability to learn the original features, especially when a significant portion of channels is shifted, resulting in the loss of valuable original information at the current timestep. To address this issue, as shown in Figure 2(c), the shifted spike features $Z$ are combined with the original input $X$ through a residual shift to insert the TS module, mitigating the information loss caused by the shifting operation. This residual shift method preserves the original features while allowing the network to capture temporal dynamics. Mathematically, the residual shift is defined as follows:

$$Z' = \alpha \odot Z + X, \tag{7}$$

where $\alpha$ is the penalty factor that constrains the temporally shifted tensor to prevent training instability or excessive spiking activity, typically set between 0.2 and 0.5. And $\odot$ denotes element-wise multiplication. This residual shift combines original and shifted spike features with a weighted factor $\alpha$ to prevent information loss, enhancing stability during training. It also alleviates the problem of learning degradation in the transmission of spiking temporal features. By integrating historical and future features through the shifting operation, this approach emphasizes specific features of the current timestep. This method ensures that while the temporal shifts incorporate features from multiple timesteps, the complete information of the original timestep is preserved, thereby minimizing the loss of information during spike transmission.

## 4. Experiments

To validate the effectiveness of the proposed TS-SNN, we evaluate its performance across four datasets, and various network architectures. First, we provide details on the datasets and implementation specifics. Next, we present extensive ablation experiments to optimize the TS module. Subsequently, we compare the performance of TS-SNN with state-of-the-art methods on static image classification tasks and event-based vision tasks. Afterward, we evaluate the generality of the TS module on Transformer-based architecture. Finally, we analyze the computational efficiency of the proposed method. More details on the datasets, hyperparameters, additional experiments are provided in the **Appendix**.

### 4.1. Experimental Setup

**Datasets.** The proposed method was evaluated on four datasets: CIFAR-10, CIFAR-100, ImageNet, and CIFAR10-DVS. CIFAR-10 and CIFAR-100 (Krizhevsky et al., 2010)

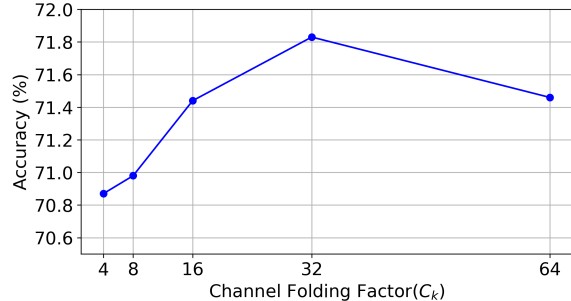

*Figure 3.* Impact of Channel Folding Factor ($C_k$) on CIFAR-100 Accuracy with ResNet20 architecture. The accuracy peaks when $C_k$ is set to 32.

are standard benchmarks for image classification, consisting of 50,000 training images and 10,000 testing images, all sized $32 \times 32$. CIFAR-10 comprises 10 classes, while CIFAR-100 contains 100 classes. ImageNet (Deng et al., 2009) is a large-scale dataset extensively used for benchmarking image classification algorithms. It includes 1.2 million training images, 50,000 validation images, and 100,000 test images, categorized into 1,000 distinct classes representing a wide range of objects. CIFAR10-DVS (Li et al., 2017b) is a neuromorphic dataset derived from the frame-based CIFAR-10 dataset using a dynamic vision sensor (DVS). It comprises 10,000 event streams, each sized $128 \times 128$, divided into 10 categories with 1,000 samples per class. We follow the convention of previous studies (Wang et al., 2023b) by splitting the dataset into training and testing sets in a 9:1 ratio.

**Implementation Details.** The entire codebase was implemented in PyTorch in this study. Experiments on CIFAR-10, CIFAR-100 and CIFAR10-DVS were conducted using an NVIDIA RTX 3090 GPU, while experiments on ImageNet were performed using 8 NVIDIA RTX 4090 GPUs. Key hyperparameters, such as the firing threshold $v_{\text{th}}$, were set to 1.0. The channel folding factor $C_k$ was set to 32, and the shift operations followed the sequence: left, right, no shift. The default value of the penalty factor $\alpha$ was 0.5. The optimization process utilized the SGD optimizer with a momentum of 0.9, an initial learning rate of 0.1, and a CosineAnneal learning rate adjustment strategy. The total number of training epochs was set to 500 for CIFAR-10, CIFAR-100, and CIFAR10-DVS, and 300 for ImageNet.

### 4.2. Ablation Studies

To optimize the TS module and gain deeper insights into its effectiveness, we conducted a series of ablation studies. These experiments were designed to isolate and evaluate the impact of various components within the TS-SNN architecture, leading to the optimal configuration.

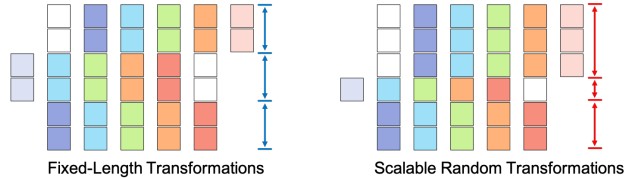

*Figure 4.* Comparison of Fixed-Length (left) and Scalable Random Transformations (right) in Temporal Feature Shift.

**Impact of Channel Folding Factor on Model Accuracy**
The principle of the TS module is to distort spike features along the temporal dimension. Different levels of distortion correspond to varying degrees of temporal spike feature integration. To facilitate block-wise feature movement, we introduce the channel folding factor, $C_k$, which controls the number of feature groups within the channels. The size of each group is determined by $C/C_k$. A larger $C_k$ results in smaller individual feature blocks, since $g_1$ and $g_2$ are randomly generated from the group, smaller groups contain less shifting information. Conversely, a smaller $C_k$ lead to richer shifting information within each group. Therefore, selecting an appropriate $C_k$ is crucial for optimal module performance. We evaluated this impact by testing the ResNet20 architecture on the CIFAR-100 dataset with a timestep of 2. The results, presented in Figure 3, show that varying $C_k$ affects performance by up to 0.96%. Ultimately, the default channel folding factor is set to 32 in our method.

**Temporal Channel Feature Shift Strategy** This study compares two strategies for the temporal feature shift method: scalable random transformation and fixed-length transformation. The primary difference between these methods lies in their approach to splitting and moving feature segments across timesteps, as depicted in Figure 4. The fixed-length transformation consistently shifts a uniform number of features, while the random transformation method varies the intervals randomly during each shift, leading to different fusion ratios between features from adjacent timesteps. To evaluate their effectiveness, we conducted experiments on the CIFAR-100 dataset under identical conditions. The results show that the fixed-length transformation achieved an accuracy of 71.24%, while the random transformation outperformed it with an accuracy of 71.63%. These findings suggest that the random transformation method provides better accuracy than the fixed-length transformation.

**Impact of TS Direction Combinations on Model Accuracy** The data presented in the Figure 5 above shows the accuracy results based on various combinations of temporal shift directions applied to different channel features. Combinations correspond to different arrangements of left, right, and present temporal shifts. The results indicate that the combination labeled as (L-R-0), which involves the con-

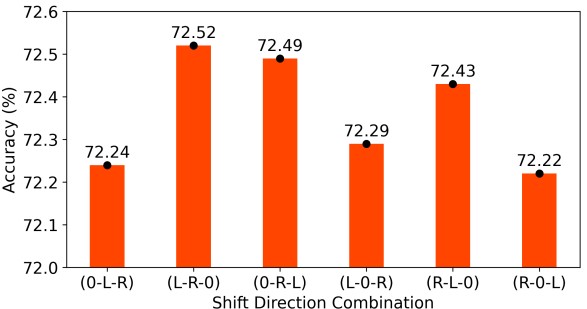

*Figure 5.* Distribution of accuracy with different TS direction combinations, with 0, L, and R representing no shift, left shift and right shift, respectively.

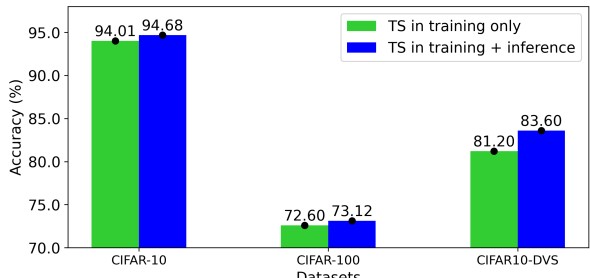

*Figure 6.* Accuracy improvement with consistent TS Module application across training and inference stages.

figuration of left-right-no shift, yields the highest accuracy at 72.52%. Consequently, this combination (L-R-0) has been default shift direction combination in all experiments conducted in this study.

**Impact of Consistent TS Module Application** We evaluated the effect of applying the TS module solely during training compared to applying it consistently across both training and inference stages. As shown in Figure 6, consistent application results in higher accuracy across all three datasets. For instance, on CIFAR10-DVS, accuracy improves from 81.20% to 83.60%. Notably, this performance gain does not increase computational costs, as the FLOPs remain unchanged in both scenarios. These results highlight the effectiveness of using the TS module in both training and inference to enhance temporal modeling while maintaining efficiency.

### 4.3. Comparisons with Other Methods

**Static Image Classification.** We evaluated our model on three static datasets: CIFAR-10, CIFAR-100, and ImageNet. The TS Module was integrated into ResNet-19 and ResNet-20 for CIFAR-10 and CIFAR-100, and into ResNet-18 and ResNet-34 for ImageNet. The results on CIFAR-10/100 are summarized in Table 1, with top-1 accuracy reported as the mean and standard deviation of 3 trials. On CIFAR-10,

*Table 1.* Comparison results with SOTA methods on CIFAR-10/100

| Methods | CIFAR-10 | | | CIFAR-100 | | |
|---|---|---|---|---|---|---|
| | Architecture | Timestep | Accuracy | Architecture | Timestep | Accuracy |
| RecDis-SNN(2022b) | ResNet-19 | 4 | 95.53% | ResNet-19 | 4 | 74.10% |
| TET(2021) | ResNet-19 | 2 | 94.16% | ResNet-19 | 2 | 72.87% |
| | ResNet-19 | 4 | 94.44% | ResNet-19 | 4 | 74.47% |
| | ResNet-19 | 6 | 94.50% | ResNet-19 | 6 | 74.72% |
| LSG(2023) | ResNet-19 | 2 | 94.41% | ResNet-19 | 2 | 76.32% |
| | ResNet-19 | 4 | 95.17% | ResNet-19 | 4 | 76.85% |
| | ResNet-19 | 6 | 95.52% | ResNet-19 | 6 | 77.13% |
| PFA(2023) | ResNet-19 | 2 | 95.60% | ResNet-19 | 2 | 76.70% |
| | ResNet-19 | 4 | 95.71% | ResNet-19 | 4 | 78.10% |
| | ResNet-19 | 6 | 95.70% | ResNet-19 | 6 | 79.10% |
| Diet-SNN(2023) | ResNet-20 | 5 | 91.78% | ResNet-20 | 5 | 64.07% |
| | ResNet-20 | 10 | 92.54% | - | - | - |
| IM-loss(2022a) | ResNet-19 | 2 | 93.85% | - | - | - |
| | ResNet-19 | 4 | 95.40% | - | - | - |
| | ResNet-19 | 6 | 95.49% | - | - | - |
| MPBN(2023c) | ResNet-19 | 1 | 96.06% | ResNet-19 | 1 | **78.71%** |
| | ResNet-19 | 2 | 96.47% | ResNet-19 | 2 | 79.51% |
| | ResNet-19 | 4 | 96.52% | ResNet-19 | 4 | 80.10% |
| | ResNet-20 | 1 | 92.22% | ResNet-20 | 1 | 68.41% |
| | ResNet-20 | 2 | 93.54% | ResNet-20 | 2 | 70.79% |
| | ResNet-20 | 4 | 94.28% | ResNet-20 | 4 | 72.30% |
| IM-LIF(2024) | ResNet-19 | 3 | 95.29% | ResNet-19 | 3 | 77.21% |
| | ResNet-19 | 6 | 95.66% | ResNet-19 | 6 | 77.42% |
| **Ours** | ResNet-19 | 1 | **96.50%** $_{\pm0.08\%}$ | ResNet-19 | 1 | 78.61% $_{\pm0.10\%}$ |
| | ResNet-19 | 2 | **96.72%** $_{\pm0.08\%}$ | ResNet-19 | 2 | **80.28%** $_{\pm0.07\%}$ |
| | ResNet-20 | 1 | **93.03%** $_{\pm0.12\%}$ | ResNet-20 | 1 | **69.02%** $_{\pm0.11\%}$ |
| | ResNet-20 | 2 | **94.11%** $_{\pm0.07\%}$ | ResNet-20 | 2 | **71.83%** $_{\pm0.10\%}$ |
| | ResNet-20 | 4 | **94.71%** $_{\pm0.08\%}$ | ResNet-20 | 4 | **73.46%** $_{\pm0.08\%}$ |

both ResNet-19 and ResNet-20 achieve the highest accuracy across all time steps. For CIFAR-100, our method surpasses others in all configurations except for ResNet-19 at timestep 1, where it lags behind MPBN. The possible reason is that at timestep 1, only the no-shift condition is present, which doesn't fully leverage dynamic temporal information, limiting its impact on performance compared to later timesteps. On the more challenging ImageNet dataset as summarized in Table 2, ResNet-34 achieves an optimal accuracy of 70.61% with a timestep of 4, marking a significant improvement. These results show that our method performs excellently with fewer time steps, demonstrating its effectiveness. The results in Table 1 demonstrate that when using a single timestep (T=1), TS theoretically should not exert any functional effect. However, we observe consistent performance improvements under this configuration. This phenomenon can be attributed to the scaling factor $\alpha$ in Equation (7). Although temporal shift operations are nullified in the spiking feature dimension with T=1, the adaptive scaling mechanism parameterized by $\alpha$ still introduces non-trivial modifications to the spiking features,

which may contribute to the observed performance gains.

**Event-based Vision classification.** To comprehensively evaluate the spatiotemporal processing capabilities of TS-SNN, we tested the model on the CIFAR10-DVS dataset, which, unlike static datasets, includes a temporal dimension. The experimental results are summarized in Table 3. Our model outperformed SOTA methods, achieving an accuracy of 83.90% with 16 timesteps, and 83.80% with only 10 timesteps, which demonstrate that TS-SNN can surpass SOTA performance while using significantly fewer timesteps. More importantly, these results highlight the superior performance of the TS module in handling longer timesteps, indicating its effectiveness in mitigating information loss over long sequences.

### 4.4. TS Module on Transformer-Based Architectures

To evaluate the generality of the TS module, we integrated it into Spikeformer-4-384, a transformer-based architecture(Zhou et al., 2022). Experiments on CIFAR-10 and

*Table 2.* Comparison results with training based SNN SOTA on ImageNet. T denotes Timestep.

| Methods | Architecture | T | Accuracy |
|---|---|---|---|
| STBP-tdBN (2021) | ResNet34 | 6 | 63.72% |
| TET (2021) | ResNet34 | 6 | 64.79% |
| RecDis-SNN (2022b) | ResNet34 | 6 | 67.33% |
| GLIF (2022) | ResNet34 | 4 | 67.52% |
| IM-Loss (2022a) | ResNet18 | 6 | 67.43% |
| Real Spike (2022c) | ResNet18 | 4 | 63.68% |
| | ResNet34 | 4 | 67.69% |
| RMP-Loss (2023b) | ResNet18 | 4 | 63.03% |
| | ResNet34 | 4 | 65.17% |
| MPBN (2023c) | ResNet18 | 4 | 63.14% |
| | ResNet34 | 4 | 64.71% |
| SEW ResNet (2021a) | ResNet18 | 4 | 63.18% |
| | ResNet34 | 4 | 67.04% |
| **Ours** | **ResNet18** | **4** | **68.18%**$_{\pm 0.18\%}$ |
| | **ResNet34** | **4** | **70.61%**$_{\pm 0.20\%}$ |

*Table 3.* Comparison results with SOTA methods on CIFAR10-DVS. T denotes timestep.

| Methods | Architecture | T | Accuracy |
|---|---|---|---|
| IM-loss(2022a) | ResNet-19 | 10 | 72.60% |
| LSG(2023) | ResNet-19 | 10 | 77.90% |
| MPBN(2023c) | ResNet-19 | 10 | 74.40% |
| MPBN(2023c) | ResNet-20 | 10 | 78.70% |
| TET(2021) | VGGSNN | 10 | 77.30% |
| IM-LIF(2024) | VGG-13 | 10 | 80.50% |
| GLIF(2022) | 7B-wideNet | 16 | 78.10% |
| STSA(2023b) | STS-Transformer | 16 | 79.93% |
| Spikeformer(2022) | Spikeformer | 16 | 80.90% |
| SEW(2021a) | SEW-ResNet | 16 | 74.40% |
| PLIF(2021b) | PLIFNet | 20 | 74.80% |
| **Ours** | ResNet-20 | 10 | **83.80%**$_{\pm 0.20\%}$ |
| | | 16 | **83.90%**$_{\pm 0.20\%}$ |

CIFAR-100 datasets demonstrate its effectiveness, as shown in Table 4. The TS module enhanced accuracy on both datasets, with a notable +0.38% gain on CIFAR-100. These results highlight its ability to integrate temporal features effectively, even in transformer-based models, demonstrating its adaptability across diverse neural network architectures.

### 4.5. Analysis of Computation Efficiency

In this section, the energy cost is estimated based on the number of operations in 45-nm technology during single-image inference. The number of multiplication-and-accumulation operations (MACs) remains constant for a given network in ANNs, while in SNNs, computations are primarily accumulation operations (ACs) triggered by incoming spikes. The energy consumption for both ANN and SNN models is based on the number of MACs and ACs(Horowitz, 2014), where each of the operation consume 4.6 pJ and 0.9 pJ, respectively(Qiao et al., 2015). The computational energy consumption across three datasets is summarized in Table 5. The method in this experiment is based on (Chen et al., 2023), where the energy consumption of SEW ResNet18 is reported to be 13.10 mJ. Notably, Under the consistent experimental conditions, our proposed TS-SNN consumes only 5.857 mJ and results for other datasets are also significantly low. This substantial reduction in energy consumption may be attributed to the sufficiently high spiking rate, demonstrating the clear energy efficiency advantage of our proposed method.

*Table 4.* Results on CIFAR-10 and CIFAR-100 with and without TS module on Transformer-based architecture.

| Methods | Accuracy | |
|---|---|---|
| | **CIFAR-10** | **CIFAR-100** |
| Spikeformer-4-384 | 95.19% | 77.86% |
| Spikeformer-4-384 with **TS** | 95.28% | 78.24% |

*Table 5.* Computational consumption for processing a single sample of CIFAR-100, ImageNet, and CIFAR10-DVS.

| Dataset | CIFAR-100 | CIFAR10-DVS | ImageNet |
|---|---|---|---|
| **Timestep** | 4 | 10 | 4 |
| **Architecture** | ResNet20 | ResNet20 | SEW ResNet18 |
| **ACs** | 141.2M | 1.23G | 1.62G |
| **MACs** | 53.99M | 514.81M | 956.4M |
| **FLOPs** | 868.36M | 8.65G | 7.29G |
| **Param** | 11.3M | 11.2M | 11.6M |
| **Energy** | 0.375mJ | 3.475mJ | 5.857mJ |

## 5. Conclusion

Building on a deep understanding of SNNs and inspired by temporal modeling and neuroscience, this paper introduces an novel Temporal Shift Module for SNNs. This module achieves a simple yet effective fusion of past, present, and future spike features within each one timestep through a shift operation. By introducing minimal computational cost, it effectively reduces the forgetting of past timestep information, enables learning from future timesteps, and

establishes robust long-term temporal dependencies as well. We conducted experiments on four datasets and performed extensive ablation studies, demonstrating that the proposed TS module significantly improves SNN accuracy. As a plug-and-play module, it shows great potential for widespread application. Additionally, energy consumption results indicate that TS-SNNs are highly energy-efficient.

## Acknowledgements

This work was supported in part by National Natural Science Foundation of China (NSFC) (62476035, 62206037, 62276230, 62271361, U24B20140), and Natural Science Foundation of Zhejiang Province (LDT23F02023F02), and State Key Laboratory (SKL) of Biobased Transportation Fuel Technology.

## Impact Statement

This paper presents work whose goal is to advance the field of Machine Learning. There are many potential societal consequences of our work, none which we feel must be specifically highlighted here.

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

# A. Appendix.

## A.1. LIF Modal

Mathematically, a LIF neuron can be represented as follows:

$$\tau \frac{du(t)}{dt} = -(u(t) - u_{\text{rest}}) + I(t), \tag{8}$$

where $\tau$ denotes the membrane time constant, $u(t)$ presents the membrane potential at the moment $t$, $u_{rest}$ is the resting potential of the neuron, and $I(t)$ is pre-synaptic input at $t$ moment.

## A.2. Datasets Details and Augmentations

**CIFAR-10.** CIFAR-10 (Krizhevsky et al., 2010) is a widely recognized benchmark dataset for image classification tasks. It consists of 50,000 training images and 10,000 test images, all sized at $32 \times 32$. The dataset contains 10 distinct classes, representing common objects such as airplanes, cars, birds, and cats. CIFAR-10 serves as a standard for evaluating the performance of image classification algorithms, offering a diverse set of visual challenges. In our approach, we apply data augmentation techniques including cropping, horizontal flipping, and cutout. Additionally, during training, we introduce random augmentations by selecting two strategies from contrast enhancement, rotation, and translation. These augmentations enhance the model's robustness and its ability to generalize across various visual scenarios.

**CIFAR-100.** CIFAR-100 (Krizhevsky et al., 2010) is an extension of the CIFAR-10 dataset, designed for more fine-grained classification tasks. It comprises 50,000 training images and 10,000 test images, all with dimensions of $32 \times 32$. The dataset features 100 classes, each belonging to one of 20 superclasses, making it a more challenging benchmark compared to CIFAR-10. The data augmentation strategy used for CIFAR-100 is consistent with that applied to CIFAR-10, providing a robust evaluation framework for models across a broader range of object categories.

**ImageNet.** ImageNet (Deng et al., 2009) is a large-scale image dataset extensively used for benchmarking image classification algorithms. It includes 1.3 million training images across 1,000 categories, along with 50,000 validation images. These classes represent a wide range of objects, including animals, vehicles, and everyday items. Compared to CIFAR-10 and CIFAR-100, ImageNet presents a larger and more complex collection of images, offering a more robust benchmark for evaluating model performance. In our experiments, we utilize the data augmentation techniques outlined in (He et al., 2016). Images are randomly cropped from either their original version or a horizontally flipped version to a size of $224 \times 224$ pixels, followed by data normalization. For testing samples, images are resized to $224 \times 224$ pixels and subject to center cropping, after which data normalization is also applied.

**CIFAR10-DVS.** CIFAR10-DVS (Li et al., 2017a) is an event-based version of the CIFAR-10 dataset, captured using a dynamic vision sensor (DVS) camera. This dataset consists of 10,000 event streams, each sized at $128 \times 128$, derived from the original CIFAR-10 images. The 10 classes in CIFAR10-DVS each contain 1,000 samples, and the dataset is split into training and testing sets with a 9:1 ratio. During preprocessing, random horizontal flips (with a probability of 0.5) are applied, followed by random selection from augmentations such as rolling, rotation, cutout, and shear. These augmentations enhance the variability and robustness of the dataset(Guan & Zhao, 2022; Guan et al., 2020).

## A.3. Experimental Setups

All code implementations were based on the PyTorch framework. The experiments were conducted on a single RTX 3090 GPU for all datasets except ImageNet, which was trained using a configuration of eight RTX 4090 GPUs. In all experiments, the SGD optimizer with a momentum of 0.9 was used, along with the CosineAnnealing learning rate adjustment strategy.

**CIFAR-10/100.** For the CIFAR-10 and CIFAR-100 datasets, the initial learning rate was set to 0.1, with a batch size of 128 and the number of training epochs set to 500. The parameter $\alpha$ was initialized at 0.5 and $C_k$ was set at 32.

**CIFAR10-DVS.** For the CIFAR10-DVS dataset, the initial learning rate was set to 0.1, with a batch size of 32 and 300 training epochs. The parameter $C_k$ was set at 32, and $\alpha$ was initialized at 0.2.

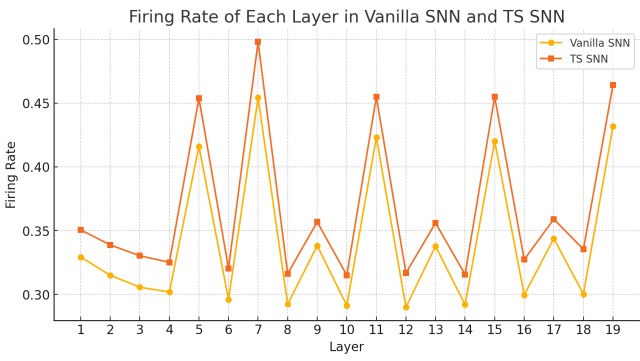

*Figure 7.* Average Firing Rate of each layer in Vanilla SNN and TS SNN

**ImageNet.** For the ImageNet dataset, the initial learning rate was set to 0.1, with a batch size of 64 and a total of 320 training epochs. The parameter $C_k$ was set at 32, and $\alpha$ was initialized at 0.2.

### A.4. Analysis of Computation Efficiency

In ANNs, each operation involves a multiplication and accumulation (MAC) process. The total number of MAC operations (#MAC) in an ANN can be calculated directly and remains constant for a given network structure. In contrast, spiking neural networks (SNNs) perform only an accumulation computation (AC) per operation, which occurs when an incoming spike is received. The number of AC operations can be estimated by taking the layer-wise product and sum of the average spike activities, in relation to the number of synaptic connections.

$$\#\text{MAC} = \sum_{l=1}^{L} (\#\text{MAC}_l) \tag{9}$$

$$\#\text{AC} = \sum_{l=2}^{L} (\#\text{MAC}_l \times a_l) \times T \tag{10}$$

Here, $a_l$ represents the average spiking activity for layer l. The first, rate-encoding layer of an SNN does not benefit from multiplication-free operations and therefore involves MACs, while the subsequent layers rely on ACs for computation. The energy consumption E for both ANN and SNN, accounting for MACs and ACs across all network layers, is given by: $E_{SNN} = \#\text{MAC}_1 \times E_{MAC} + \#\text{AC} \times E_{AC}$ and $E_{ANN} = \#\text{MAC} \times E_{MAC}$.

Based on previous studies in SNN research (Yao et al., 2023; Chakraborty et al., 2021), we assume that the operations are implemented using 32-bit floating-point (FL) on a 45 nm 0.9V chip (Horowitz, 2014), where a MAC operation consumes 4.6 pJ and an AC operation consumes 0.9 pJ. This comparison suggests that one synaptic operation in an ANN is roughly equivalent to five synaptic operations in an SNN. It is important to note that this estimation is conservative, and the energy consumption of SNNs on specialized hardware designs can be significantly lower, potentially reduced by up to 12× to 77 fJ per synaptic operation (SOP) (Qiao et al., 2015).

### A.5. Analysis of Firing Rates

Firing rates play a crucial role in biological neural networks, where they are used to encode information, forming the basis of neural communication and signal processing. In computational models like SNNs, the firing rate is a key metric for simulating and analyzing neuronal dynamics, providing insights into the role of neurons in complex behaviors and information processing tasks. The firing rate quantifies the activity level of a neuron over a specified period, serving as a vital indicator of the neuron's responsiveness to input stimuli—higher firing rates typically signify stronger responses.

In our analysis, we compare the firing rates of the proposed TS-SNN with a vanilla SNN model, as summarized in Figure 7. Firing rates were recorded across four timesteps, with the average firing rate for each layer calculated based on these timesteps. This comparison highlights the differences in neuronal activity between the two models, providing insights into their respective processing efficiencies.

In the context of the overall network, the spike firing rate of the vanilla SNN is 33.06%, while that of the TS-SNN is 35.61%. This suggests that the TS module enhances model performance without substantially increasing spike output, thereby preserving the inherent sparsity of the original SNN.

