# OpenReview forum: "TS-SNN: Temporal Shift Module for Spiking Neural Networks"
_ICML.cc/2025/Conference — ICML 2025 poster_

### Official Review · Reviewer_x8Ri · 2025-03-07

**Overall Recommendation:** 4

**Summary:**

The paper introduces the Temporal Shift module for Spiking Neural Networks, by utilizing a Temporal Shift operation, the model integrates past, present, and future spike features within a single timestep, aiming to improve the temporal dynamics of SNNs. The key advantage of TS-SNN lies in its ability to model temporal dependencies efficiently, enhancing performance with minimal increasing computational costs. Experimental results across multiple benchmark datasets demonstrate that TS-SNN outperforms SOTA models, achieving high accuracy with fewer timesteps. The approach is energy-efficient, aligning well with the goals of neuromorphic computing.

**Claims And Evidence:**

The claims regarding the efficiency and performance of the TS-SNN method are well-supported by experimental evidence with convincing comparisons against existing methods on benchmark datasets, demonstrating superior performance in terms of accuracy and energy consumption.

**Essential References Not Discussed:**

While the paper does a good job of citing important related works, some recent developments in spiking neural networks and temporal modeling might be relevant but aren’t discussed in detail. For example, a more thorough review of the intersection between SNNs and transformer architectures could have been beneficial.

**Experimental Designs Or Analyses:**

The experimental design is sound. The authors conduct extensive ablation studies to analyze the effectiveness of various components of the TS-SNN, such as the channel folding factor and the temporal shift strategy. The results are robust and show the effectiveness of the method across different architectures. The experiments on energy consumption are particularly valuable, highlighting the energy efficiency of the approach compared to other models.

**Methods And Evaluation Criteria:**

Yes. The Temporal Shift module is well-suited for enhancing the temporal dynamics of SNNs, and the benchmark datasets used are standard for evaluating image classification and event-based vision tasks. The use of CIFAR-10, CIFAR-100, ImageNet, and CIFAR10-DVS datasets ensures that the model is tested under varied conditions, demonstrating its robustness across different types of data.

**Other Comments Or Suggestions:**

Formatting improvement: some equations could be better aligned.

**Other Strengths And Weaknesses:**

The paper’s strength lies in its originality, particularly in the introduction of the Temporal Shift module for SNNs, which enhances temporal modeling with minimal computational overhead. The experimental results are compelling, demonstrating the ability of the TS-SNN to outperform existing models with fewer timesteps. However, a more thorough discussion of potential limitations (e.g., situations where the temporal shift might be less effective) would provide a more balanced perspective. Additionally, in Tab. 1, the performance of ResNet-19 with a timestep of 1 on CIFAR-100 is lower than MPBN, while it outperforms MPBN on CIFAR-10. The paper does not provide a clear explanation for this discrepancy.

**Questions For Authors:**

Could the authors provide an explanation for the lower performance of ResNet-19 on CIFAR-100 compared to MPBN when the timestep is set to 1, as shown in Table 1?

**Relation To Broader Scientific Literature:**

As the authors mentioned, the paper builds on existing work in temporal modeling and relates their approach to earlier methods such as the Temporal Shift Module and 3D CNNs. This paper introduces the Temporal Shift to SNNs, highlighting the novelty of incorporating spatial-temporal features into SNNs. The comparison with SOTA methods further demonstrates the significance of this contribution.

**Theoretical Claims:**

The theoretical claims regarding the TS module’s ability to efficiently integrate Spatio-temporal features through shifting operations are clearly explained. There are no formal proofs provided for these claims, but approach is based on well-understood principles in deep learning and features of SNNs and the experimental results substantiate the theory.

---

> ### Author Rebuttal · Authors · 2025-04-01
>
> ## Questions for Authors
>
> **Performance Discrepancy for ResNet-19 on CIFAR-100 (Timestep=1)**
>
> Thank you for pointing out this discrepancy. As you correctly observed, the performance of our TS-SNN with ResNet-19 on CIFAR-100 under a single timestep lags slightly behind MPBN, whereas it outperforms MPBN on CIFAR-10. This behavior can be attributed to the nature of the **Temporal Shift (TS) operation** when T=1. In this case, there are no past or future information to reference, and the shift operation defaults to zero-padding. The resulting update becomes a residual connection:
>
> $Z' = \alpha Z + X$.
>
> As a result, the TS operation brings limited benefit when T=1, and the model's performance largely relies on the backbone architecture and input statistics.
> Notably, CIFAR-100 is a more challenging dataset with fewer samples per class, which amplifies the limitations of reduced temporal modeling under T=1.
> This observation demonstrates that the primary advantage of our method arises from the effective feature fusion brought by the temporal shift operations across multiple timesteps.
> We will include this clarification in the camera-ready version of the paper.
>
> ---
> ## Weakness
>
> ### 1. Lack of Discussion on Limitations
>
> We appreciate your suggestion. We will include two primary limitations of our method:
>
> - **Restricted Effectiveness at Low Timesteps**
>   As mentioned above, when T=1, the TS module cannot utilize temporal context, limiting its effectiveness. The model reverts to near-baseline behavior, and performance gains are minimal or inconsistent.
>
> - **Memory Requirements**
>   The TS operation assumes access to spike outputs from both past and future timesteps within a local window. On neuromorphic hardware, this may require additional memory and access latency to buffer or store spike states across timesteps. While this overhead is generally modest compared to the savings in computation and energy (since spike buffers are lightweight), it could become a bottleneck for highly constrained or streaming systems.
> We plan to explicitly discuss these limitations and potential mitigations in the camera-ready version.
>
> ---
>
> ### 2. Performance Discrepancy in Table 1
>
> Please see our response above under ##Questions for Authors## for a detailed explanation.
>
> ---
>
> ## Other Comments and Suggestions
>
> **Formatting Issues**
>
> We appreciate your feedback on formatting. We have revised the LaTeX alignment of all equations to ensure consistency—either center alignment or equation numbering, depending on context. This significantly improves the visual quality and readability of the manuscript.
>
> ---
> We sincerely thank the reviewer for their thoughtful and helpful suggestions. We will incorporate all the suggested changes into the camera-ready version of the paper.

---

> > ### Comment · Reviewer_x8Ri · 2025-04-05
> >
> > Thanks for the response. My concerns have been addressed.

---

### Official Review · Reviewer_cLUM · 2025-03-10

**Overall Recommendation:** 3

**Summary:**

This paper introduce a Temporal Shift module for SNN called TS-SNN, which enhances the ability of SNNs for temporal information. The TS-SNN consists of two parts. The first part, Temporal Shift module (TS), divides the spike output matrix into C_k groups in the channel dimension, and divides each group into three parts: left shift, right shift, unchanged shift in the time dimension, respectively. The second parts is a residual connection that adds the shifted spike matrix to the original spike matrix according to a ratio $\alpha$. The authors conducted ablation experiments to find best parameters setting of the TS module. And the comparative experimental results show that TS-SNN achieves excellent results on both static datasets and neuromorphic datasets.

**Claims And Evidence:**

The authors claim that the TS module can improve the ability of SNN to extract temporal information, but they have not theoretically or experimentally demonstrated their claim.

**Essential References Not Discussed:**

NA

**Experimental Designs Or Analyses:**

Yes, I have checked the author's experimental design, and the experiment is relatively reasonable. However, the authors used larger training epochs, which makes me doubt whether the their method will cause slow training convergence.

**Methods And Evaluation Criteria:**

The TS module, if used as a training method, can improve the robustness of SNNs to the noise (spike shift or spike loss) to a certain extent. This method has a certain influence in the SNN field.

**Other Comments Or Suggestions:**

Figures 5 and 6 should provide the baseline without TS-SNN method.

**Other Strengths And Weaknesses:**

Strength:
1. The method proposed in this paper is simple and effective, achieving SOTA performance on both static datasets and neuromorphic datasets.
2. The writing of the paper is concise and clear.

Weakness:
1. The authors claims that TS-SNN can enhance the ability of SNN to extract temporal information, but the paper lacks relevant theoretical analysis and experimental verification as to why it is effective.
2. The TS method requires information about the spike output at previous and subsequent moments. However, on the neuromorphic chips, it difficult to obtain the spike information at different moments (requires additional memory and cost to record the spikes).
3. The setting of $\alpha$ is different for different datasets, and the paper does not provide the setting rule for $\alpha$, which may reduce the practicality of the method.
4. The authors used more training epochs in the experiment part, exceeding the common settings in other SNN works.

**Questions For Authors:**

In Table 1, the author demonstrates the results under timestep is 1, where the SNN does not use any temporal information. In this situation, how does the TS method operation, and does the TS method improve training performance?

**Relation To Broader Scientific Literature:**

NA

**Theoretical Claims:**

The authors did not provide theoretical proof for their methods.

---

> ### Author Rebuttal · Authors · 2025-03-31
>
> Thank you for your detailed review and constructive feedback. We address your points as follows:
>
> ---
> ## Questions for Authors
>
> **How TS operation when timestep = 1**
>
> As you correctly observed, the performance of our TS-SNN with ResNet-19 on CIFAR-100 under a single timestep lags slightly behind MPBN, whereas it outperforms MPBN on CIFAR-10. This behavior can be attributed to the nature of the **TS operation** when T=1. In this case, there are no past or future information to reference, and the shift operation defaults to zero-padding, leaving only the residual connection  $Z' = \alpha Z + X$ active. Although this residual fusion can subtly affect feature extraction, it does not consistently enhance model stability. This observation demonstrates that the primary advantage of our method arises from the effective feature fusion brought by the temporal shift operations across multiple timesteps.
> We will include this clarification in the camera-ready version of the paper.
>
> ---
> ## Weakness
>
> ### 1. Theoretical Analysis and Experimental Verification
>
> We appreciate your observation.
> In the camera-ready version, we will incorporate theoretical insights that will substantially strengthen the paper, including:
>
> - **Expansion of the Temporal Receptive Field**
> By shifting features forward and backward, the TS module enables neurons to access information from adjacent timesteps, which is effectively allowing each neuron to "see" more temporal context. This operation expands the temporal receptive field from \(d\) to \(2d + 1\) without incurring extra parameters or computational cost.
>
> - **Increase in Mutual Information and Entropy**
> The TS module fuses inputs ${X_{t-1}, X_t, X_{t+1}}$ to compute $S_t$, thereby increasing its temporal entropy $H(S_t)$. Under stationary input conditions, the mutual information $I(S_t; X_{t-1:t+1})$ exceeds $I(S_t; X_t)$, demonstrating enhanced temporal context modeling.
>
> ---
> ### 2. Implementation on Neuromorphic Hardware
>
> Your concern that the TS module may incur additional memory costs on neuromorphic chips is reasonable. However, relative to the significant energy efficiency gains demonstrated by our approach, the extra storage requirement is minimal. The hardware optimization strategies will be promising in future work to further mitigate this cost in practical deployments.
>
> ---
> ### 3. Setting of the $\alpha$ Parameter
>
> We appreciate your insightful observation regarding the $\alpha$ parameter. In our implementation details, we mistakenly stated that the initial value of $\alpha$ is 0.5 for all datasets. The correct settings, as detailed in the supplementary material, are as follows: for CIFAR-10 and CIFAR-100, the initial $\alpha$ is 0.5; for ImageNet and CIFAR10-DVS, the initial $\alpha$ is 0.2. We will correct this error in the camera-ready version.
>
> Furthermore, we specified in the manuscript that the experimental range for$\alpha$ is between 0.2 and 0.5. Our experiments validated the effectiveness of this range across different datasets. We found that the initial value of $\alpha$ is critical—if it is set above 0.5, training tends to collapse. Therefore, we recommend tuning $\alpha$ as a hyperparameter within the 0.2–0.5 range based on the specific dataset.
>
> ---
> ### 4. Training Epochs and Convergence
>
> Although our experiments on CIFAR-10 and CIFAR-100 used 500 epochs, we observed that competitive performance can be achieved with as few as 200 epochs—the performance difference is within 0.4% as shown in the table below.
> We chose 500 epochs to align with some current SOTA settings and to ensure robust convergence.
> In the camera-ready version, we will include this experiments to clarify this.
>
> | Archi | T | CIFAR10 200 | CIFAR10 500 | ↑200→500 | ↑200→BL | ↑500→BL | CIFAR100 200 | CIFAR100 500 | ↑200→500 | ↑200→BL | ↑500→BL |
> |-|-|-|--|--|--|-|-|-|-|-|-|
> | | |\% |\%|\%|\%|\%|\%|\%|\%|\%|\%|
> | R-19 | 1 | 96.27|96.50|.23 | .21| .44| 78.24 |78.61|.37 | -.47| -.10 |
> | R-19 | 2 | 96.54|96.72|.18 | .07| .25 | 79.99|80.28|.29 | .48| .77|
> | R-20 | 1 | 92.68|93.03|.35 | .46| .81| 68.90 |69.02|.12 | .49| .61|
> | R-20 | 2 | 93.99|94.11|.12 | .45| .57| 71.58|71.83|.25 | .79| 1.04|
> | R-20 | 4 | 94.39|94.71|.32 | .11| .43| 73.07|73.46|.39 | .77| 1.16|
>
> ---
> ## Other Comments and Suggestions
>
> Thank you for your suggestion regarding Figures 5 and 6. In the camera-ready revision, we will update these figures to include baseline data (i.e., models without the TS module), clearly demonstrating the performance improvements attributable solely to the TS module.
>
> ---
> We greatly appreciate your valuable advice, which will help enhance the quality and clarity of our work.

---

> > ### Comment · Reviewer_cLUM · 2025-04-05
> >
> > Thanks for your response, I have raised my score. However, for the Theoretical Analysis part, I personally speculate that the temporal shift operation makes the training gradient of SNN more accurate, but whether my speculation or the authors theoretical insights, more detailed verification experiments are still needed.

---

### Official Review · Reviewer_bUca · 2025-03-12

**Overall Recommendation:** 3

**Summary:**

Research related to spiking neural networks (SNNs) has received increasing attention, but it is still a challenge to strike a balance between time steps and low energy consumption. In this paper, we introduce the Time Shift Module for Spiking Neural Networks (TS-SNN), which integrates past, present, and future spike features within a single time step through a simple but effective shift operation. The residual combination method prevents information loss by integrating shifted and original features. The model achieves optimal performance on multiple datasets.

**Claims And Evidence:**

Yes, the claims made in the submission are supported by clear and convincing evidence.

**Essential References Not Discussed:**

I think the paper has cited enough relevant literatures.

**Experimental Designs Or Analyses:**

The authors have experimentally validated the proposed method in terms of both model performance and energy consumption analysis, and the results demonstrate optimal performance and lower power consumption.

**Methods And Evaluation Criteria:**

The time-shift module moves some channels forward by a +1 operation and others backward by a -1 operation, while the rest of the channels remain unchanged along the time dimension. By integrating the spike features of different time steps, TS-SNN effectively mitigates the effect of the loss of the information of the initial time step and solves the problem of the insufficient feature extraction of the end time step. The proposed method can effectively reduce the forgetting of past time-step information, realize the learning of future time-steps, and establish robust long-term temporal dependencies.

**Other Comments Or Suggestions:**

None

**Other Strengths And Weaknesses:**

None

**Questions For Authors:**

Due to the lightweight nature of the TS module, it has the potential to be widely used in other tasks besides classification tasks as well, can the authors give some validation?

**Relation To Broader Scientific Literature:**

As a plug-and-play module, the TS module has great potential for a wide range of applications

**Theoretical Claims:**

Not applicable. This paper does not involve complex theoretical proofs.

---

> ### Author Rebuttal · Authors · 2025-04-01
>
> Thank you for this insightful question. Indeed, one of the advantages of the Temporal Shift module is its lightweight and general design, making it applicable beyond classification.
>
> While our current work focuses on image classification due to space constraints and the need for standardized comparisons with prior SNN models, we agree that the TS module has strong potential in other tasks such as object detection, semantic segmentation, or event-based video understanding.
>
> In future work, we plan to extend TS-SNN to such domains. Additionally, the TS module is designed to be plug-and-play and architecture-agnostic, so it can be seamlessly integrated into models for these tasks with minimal modifications.
>
> We appreciate the suggestion and will include a discussion of this in the camera-ready version.

---

### Decision · Program_Chairs · 2025-05-01

**Decision:**

Accept (poster)

**Comment:**

This paper proposes TS-SNN, a lightweight and effective method to enhance temporal feature extraction in Spiking Neural Networks (SNNs) via a novel Temporal Shift module. The module shifts spike features across time and integrates them with a residual connection, significantly improving the temporal receptive field while maintaining low computational and memory overhead.

### **Strengths**
- **Technical novelty and simplicity**: The proposed TS module is architecture-agnostic, lightweight (introducing only one learnable parameter), and easy to integrate, making it a practical contribution for the SNN community.
- **Strong empirical results**: The method demonstrates state-of-the-art performance on several benchmark datasets, including CIFAR-10, CIFAR-100, ImageNet, and CIFAR10-DVS. It also maintains energy efficiency with fewer timesteps, aligning well with the goals of neuromorphic computing.
- **Thorough experimentation**: The paper includes ablation studies and comparisons with strong baselines. Reviewers appreciated the experimental rigor, including supplementary analysis on energy consumption and hyperparameter sensitivity.
- **Clarity and writing**: The manuscript is well-written, and the authors’ rebuttals are detailed and constructive. They committed to incorporating several improvements in the camera-ready version, including additional theoretical explanations, clearer experimental settings, and better formatting.

### **Weaknesses**
- **Lack of theoretical depth**: While the authors provided some theoretical insights in the rebuttal (e.g., expanded temporal receptive field, mutual information increase), the initial manuscript lacked formal theoretical analysis or verification experiments to support the core claim that temporal shifting improves gradient flow or learning dynamics in SNNs.
- **Hardware realism**: Reviewers raised valid concerns about the feasibility of implementing the temporal shift operation on neuromorphic hardware due to the need for spike access across timesteps, which may incur memory and latency costs. Although the authors acknowledged this and argued the overhead is modest, the issue remains relevant for real-world deployment.
- **Hyperparameter generality**: The method requires tuning the residual fusion ratio (α), and the appropriate setting differs by dataset. The paper did not provide a principled guideline for selecting this parameter, which could impact reproducibility and generalization.
- **Limited effectiveness at T=1**: Reviewers noted and the authors confirmed that the TS module has little benefit when the timestep is set to 1, as the operation degenerates into residual fusion with zero-padding. This limits the method’s applicability in extremely low-timestep regimes.

### **Rebuttal Assessment**
The authors responded thoroughly to all reviewer concerns. They clarified the working mechanism of TS at T=1, provided additional experimental results showing performance is not overly dependent on training epochs, and outlined theoretical intuitions for the method’s effectiveness. Their planned additions (e.g., discussion of limitations, parameter clarification, formatting improvements) will significantly strengthen the final version.